# Design, Preparation, Characterization and Evaluation of Five Cocrystal Hydrates of Fluconazole with Hydroxybenzoic Acids

**DOI:** 10.3390/pharmaceutics14112486

**Published:** 2022-11-17

**Authors:** Hongmei Yu, Baoxi Zhang, Meiju Liu, Wenhui Xing, Kun Hu, Shiying Yang, Guorong He, Ningbo Gong, Guanhua Du, Yang Lu

**Affiliations:** 1Beijing Key Laboratory of Polymorphic Drugs, Institute of Materia Medica, Chinese Academy of Medical Sciences, Peking Union Medical College, Beijing 100050, China; 2Beijing City Key Laboratory of Drug Target Identification and Drug Screening, Institute of Materia Medica, Chinese Academy of Medical Sciences, Peking Union Medical College, Beijing 100050, China

**Keywords:** fluconazole, cocrystal, hydrate, stability, dissolution

## Abstract

To modulate the physicochemical properties of fluconazole (FLZ), a multifunctional antifungal drug, the crystal engineering technique was employed. In this paper, five novel cocrystal hydrates of FLZ with a range of phenolic acids from the GRAS list, namely, 2,4-dihydroxybenzoic acid (24DHB), 3,4-dihydroxybenzoic acid (34DHB, form I and form II), 3,5-dihydroxybenzoic acid (35DHB), and 3,4,5-trihydroxybenzoic acid (345THB) were disclosed and reported for the first time. Crystals of these five hydrates were all obtained for single-crystal X-ray diffraction (SCXRD) analysis. Robust (hydroxyl/carboxyl) O−H. . . N_arom_ hydrogen bonds between acids and FLZ triazolyl moiety were observed to be dominant in guiding these crystal forms. The water molecule plays the role of supramolecular “linkage” in the strengthening and stabilization of these hydrates by interacting with FLZ and acids through O−H. . . O hydrogen bonds. In particular, the formation of FLZ−34DHB−H_2_O (1:1:1) significantly reduces hygroscopicity and hence improves the stability of FLZ, the latter of which is unstable and easily transforms into its monohydrate form. Increased initial dissolution rates were observed in the obtained cocrystal forms, and an enhanced intrinsic dissolution rate was obtained in FLZ−35DHB−H_2_O (1:1:1) in comparison with commercialized FLZ form II.

## 1. Introduction

Fluconazole, [FLZ, 2-(2,4-difluorophenyl)-1,3-di(1H-1,2,4-triazol-1-yl)propan-2-ol, Figure 1], a bis-triazole multifunctional antifungal drug used in the prevention and treatment for superficial and systemic fungal infections, was firstly reported by Richardson in 1983 [1]. To our knowledge, solid forms display variability and exist in polymorphic forms, which refer to crystalline and amorphous forms as well as solvates and hydrates [2,3,4,5,6,7,8]. FLZ has been found to exhibit at least seven anhydrous polymorphs, including three main earlier reported polymorphs, namely I, II, and III by Lo et al. [9], Gu and Jiang [10], Dash and Elmquist [11], Richardson et al. [12], and four more recent ones reported by Karanam et al. [13], namely, IV, V, VI, and VII. Among the three earlier found polymorphs of FLZ, form I (CSD refcode IVUQOF) is the most stable, form II has been commercialized, and form III is the most soluble but metastable. An ethyl acetate solvate and two monohydrate forms of FLZ were reported by Caira [14], Basford and Cameron [15]. The marketed FLZ used in this study was identified as “polymorph I” by Gu and Jiang [10], “form II” by Dash and Elmquist [11], “polymorph 2” by Karanam [13], or “anhydrate form I” by Alkhamis et al. [14]. Usually, there are two main challenges associated with commercialized FLZ form II, characterized by powder X-ray diffraction (PXRD) but not by single-crystal X-ray diffraction (SCXRD). FLZ form II suffers from stability issues due to hygroscopic tendency, which easily transforms into its monohydrate form (CSD refcode IVUQIZ). It was reported to exhibit a poor aqueous solubility ranging from 2 to 8 mg mL^−1^ [16,17,18]. In this respect, it is of paramount importance to modulate the physicochemical properties of this widely prescribed drug in terms of stability and aqueous solubility.

In the past few decades, crystal engineering and supramolecular synthon principles have emerged as useful strategies in the pharmaceutical industry, with the ability to modulate the physicochemical properties (melting point, hygroscopicity, mechanical properties, solubility, dissolution rate, and bioavailability) of active pharmaceutical ingredients (APIs), as well as to provide intellectual property implications [19,20,21,22,23,24]. Being a weak base (p*K*a value of 1.76 for its conjugate acid [25]), salt could only be formed in the presence of very strong acids, while with weak organic acids cocrystals are more likely to be constructed. Unlike salts, in which proton transfer occurs from the acidic moiety to the basic moiety, the individual components in cocrystals are neutral and stabilized by non-ionic interactions, most often hydrogen bonds. In this respect, cocrystals are amenable even for those APIs that lack ionizable functional groups and can be used as an intriguing alternative to salts. Researchers have seen potential in improving solubility or stability properties of FLZ by employing a crystal engineering strategy and, to date, two organic acid salts, some cocrystals with carboxylic acids, and one salt cocrystal have already been developed [18,25,26,27,28,29].

The purpose of this work was to design pharmaceutical cocrystals of FLZ to modulate its stability and solubility, motivated by current advances in supramolecular crystal engineering. Herein, a range of pharmaceutical-acceptable phenolic acid nutraceuticals have been selected as CCFs (Figure 1) in the cocrystallization of FLZ, which successfully resulted in five cocrystal monohydrates, FLZ−24DHB−H_2_O (1:0.5:1), FLZ−34DHB−H_2_O (form I, 1:0.5:1; form II, 1:1:1), FLZ−35DHB−H_2_O (1:1:1), FLZ−345THB−H_2_O (1:1:1). The crystal structures and intermolecular interactions of these cocrystal hydrates were comprehensively elucidated by single-crystal X-ray diffraction (SCXRD). Complementary solid-state characterizations of powder X-ray diffraction (PXRD), Fourier transform infrared spectroscopy (FT−IR), and differential scanning calorimetry/thermogravimetric analysis (DSC/TGA) also support our study to confirm molecular assemblies.

## 2. Materials and Methods

### 2.1. Materials

FLZ raw material (purity: 98%, form II identified by PXRD [14]) was purchased from Wuhan Yinhe Chemical Co., Ltd. (Wuhan, China). FLZ standard product (purity: 99.8%) was purchased from National Institutes for Food and Drug Control (Beijing, China). Organic acids: 24DHB (98%), 34DHB (98%), 35DHB (98%), and 345THB (98%) were purchased from Beijing Chemical Reagent Co. (Beijing, China). All analytical grade solvents were purchased from Sigma Aldrich (St. Louis, MO, USA) and were used without further purification.

### 2.2. Sample Preparations

Syntheses of FLZ cocrystal powders were performed by the liquid-assisted grinding method. The specific stoichiometric ratios of FLZ and the given hydroxybenzoic acids were placed in a mortar. FLZ (612.54 mg, 2 mmol) with 24DHB (154.12 mg, 1 mmol), FLZ (306.27 mg, 1 mmol) with 34DHB (154.12 mg, 1 mmol), and FLZ (306.27 mg, 1 mmol) with 35DHB (154.12 mg, 1 mmol) were accurately weighed and ground by a pestle for 20–25 min accompanied by 5–6 drops of acetone at 25 °C. Then, powders of FLZ–24DHB–H_2_O (1:0.5:1), FLZ–34DHB–H_2_O (1:1:1), and FLZ–35DHB–H_2_O (1:1:1) were obtained successfully. The resulting powders were air-dried to remove any trace amount of solvent and collected for solid-state characterizations (PXRD, DSC, TG, FT–IR), stability, and dissolution evaluations.

A slow evaporation method was applied to prepare the single crystals of FLZ cocrystals. Powders of FLZ–24DHB–H_2_O (1:0.5:1), FLZ–34DHB–H_2_O (1:1:1), FLZ–35DHB–H_2_O (1:1:1), and FLZ–345THB–H_2_O (1:1:1) weighing 200 mg were dissolved in 10 mL ethanol: acetone (*v*:*v*, 1:1) solution, whereas FLZ–34DHB–H_2_O (1:0.5:1) were dissolved in acetonitrile solution at 25 °C, followed by sonication until complete dissolution. The resulting clear solutions were stirred at a speed of 300 rpm for 6 h and filtered, and the subsequent evaporation process was allowed to take place. Corresponding single crystals of FLZ–24DHB–H_2_O (1:0.5:1), FLZ–34DHB–H_2_O (1:1:1), FLZ–35DHB–H_2_O (1:1:1), FLZ–345THB–H_2_O (1:1:1), and FLZ–34DHB–H_2_O (1:0.5:1) suitable for SCXRD analysis were harvested under ambient conditions (20–25 °C) within 7–15 days.

### 2.3. Single-Crystal X-ray Diffraction (SXRD) Analysis

The SCXRD data were collected on a Micromax 002+ system equipped with a graphite monochromator using *Cu* Kα radiation source (*λ* = 1.54184 Å). Data reduction was performed using the CrysAlisPro program. Structure solutions were executed using intrinsic phasing in SHELXT-2018-2 [30], and refinements were performed by full-matrix least-squares on *F*^2^ using the SHELXL program [31], both implemented in the Olex2 (version 1.5, Durham University, Durham, UK) software. Non-hydrogen atoms were refined with anisotropic displacement parameters. Hydrogen atoms bonded to nitrogen and oxygen were located from difference Fourier maps and refined freely, while other hydrogens were placed at their geometrically calculated positions using a riding model. The programs Mercury (version 2020.3.0, Cambridge Crystallographic Data Center, Cambridge, UK) and Olex2 were employed to prepare artwork representations and packing diagrams.

### 2.4. Powder X-ray Diffraction (PXRD) Analysis

PXRD experiments were conducted on a Rigaku D/max−2550 X-ray diffractometer (*Cu* Kα radiation, *λ* = 1.54178 Å), equipped with a graphite monochromator, operated at 40 kV and 150 mA. The measurements were performed over the 2*θ* range of 3–40° with a step size of 0.02° at a constant scan rate of 8°/min. The data were further analyzed using Jade 6.5 software (Materials Data, Livermore, CA, USA). The simulation of PXRD patterns based on the SCXRD data was carried out using the Mercury program.

### 2.5. Thermal Analysis

DSC measurements were performed on a Mettler Toledo DSC1 Instrument (Greifensee, Switzerland). Samples weighing 3–5 mg were placed in an aluminum pan and covered with a perforated lid. The samples were heated from 30 °C to 200 °C at a scan rate of 10 °C min^−1^ under the air gas.

TG experiments were performed using a Mettler−Toledo DSC/TGA1 STARe system (Greifensee, Switzerland) for the measurements of the TG signals. The accurately weighed (8–10 mg) samples were added to an aluminum crucible under dry nitrogen gas (50 mL min^−1^). The heating was performed in the temperature range of 30–230 °C at a rate of 10 °C min^−1^.

### 2.6. Fourier Transform Infrared Spectroscopy (FT−IR) Analysis

The FT−IR spectra were collected on a Nicolet FT−IR equipped with an ATR device in the range from 4000 to 650 cm^−1^ at a resolution of 4 cm^−1^ at 16 scans under ambient conditions.

### 2.7. Stability Test

The accelerated stability [32] tests of FLZ and its cocrystal hydrates were carried out in a drug stability test instrument (SHH−150SD) under accelerated storage conditions, high temperature (60 ± 1 °C), high humidity (90 ± 5%, 25 °C), and illumination (4500 ± 500 l×, 25 °C). Approximately 50 mg of powdered samples were stored under the three test conditions and measured at regular time intervals (0, 5, and 10 days) by PXRD.

### 2.8. Apparent Powder and Intrinsic Dissolution Experiments

The powder dissolution rate of FLZ and its cocrystal hydrates were evaluated on an RC12AD dissolution apparatus (Tianjin TIANDA TIANFA−pharmaceutical testing instrument manufacturer), equipped with an RZQ−12D auto sampling station. For all the powder dissolution experiments, accurately weighed sample powders containing equivalent to 600 mg of FLZ were placed in the basket into dissolution vessels containing ultrapure water used as the dissolution medium (600 mL) at 37 ± 0.2 °C with a rotation speed of 100 rpm. Sampling was conducted by withdrawing 1 mL aliquot of the sample at time intervals of 5, 10, 15, 20, 30, 45, 60, 90, and 120 min from the vessel and filtered through a 0.45 μm syringe tip filter, and the aliquots were assayed on a high-performance liquid chromatography (HPLC) system. The measurements of the powder dissolution rate were repeated in triplicate.

Intrinsic dissolution rate (IDR) experiments of FLZ and its cocrystal hydrates were carried out in the ultrapure water and continued for up to 5 h on an RC12AD dissolution apparatus. Accurately weighed solids equivalent to 200 mg of FLZ were taken in the intrinsic attachment and compressed under a pressure of 200 kg for 1 min to provide a flat surface on one side. The resulting disk has a flat surface of 0.5 cm^2^. Then, the disk was dipped into a vessel containing 900 mL ultrapure water at 37 ± 0.2 °C, with the paddle rotating at 100 rpm. At time intervals (60, 75, 90, 120, 150, 180, 210, 240, 270, and 300 min), 1 mL of the dissolution medium was withdrawn manually. After filtration through a 0.45 μm syringe tip filter, the aliquots were assayed on an HPLC system. Each experiment was repeated in triplicate.

A HPLC instrument was used to quantify the concentration of FLZ in the samples, which is defined as the solubility of FLZ and its multi-component forms. The HPLC (Agilent Technologies, Santa Clara, USA) system was equipped with a UV detector and an Agilent Eclipse XDB−C18 (250 mm × 4.6 mm, 5 μm) column with the temperature set to 30 °C. The mobile phase consisted of acetonitrile and water (25:75, *v*:*v*) with a flow rate of 1.0 mL min^−1^. The injection volume was 10 μL and the detection wavelength was set to 261 nm. The solubility of FLZ was calculated based on the external standard method. No overlap between peaks for FLZ or any CCFs was observed.

## 3. Results and Discussion

### 3.1. Cocrystal Design

In the view of the crystal engineering aspect [33,34], the (hydroxyl/carboxyl) O–H . . . N_arom_ heterosynthon exhibits a high persistence in the Cambridge Structural Database (CSD), and aliphatic and aromatic carboxylic acids are promising CCFs that can be used to cocrystallize with nitrogen heterocycle compounds like FLZ. The two triazolyl groups in FLZ are good hydrogen bond acceptors and are likely to assemble supramolecular assemblies with complementary functionalities, such as hydroxyl and carboxylic acid groups. In addition, in the structures of reported FLZ cocrystals, O−H . . . O involving the hydroxyl group of FLZ and O−H . . . N associated with triazlyl N atoms of FLZ are two strong hydrogen bonds driving the cocrystal formation. The construction of (hydroxyl/carboxyl) O–H . . . N hydrogen bonds would therefore be expected by introducing di- and tri-hydroxybenzoic acids as CCFs in the pharmaceutical cocrystallization of FLZ.

It should also be mentioned that phenolic hydroxybenzoic acids are pharmaceutically acceptable compounds belonging to the generally regarded as safe (GRAS) chemical list released by the Food and Drug Administration (FDA), to be utilized in drug formulations [35]. Furthermore, Z. Sroka and W. Cisowski [36] have investigated and claimed that these phenolic hydroxybenzoic acids possess antioxidant properties based on hydrogen peroxide (H_2_O_2_) and 1,1-diphenyl-2-picrylhydrazyl radical (DPPH) free-radical scavenging capacities. The strong hydrogen bond donating ability together with potential health benefits make them good choices in cocrystallization.

Surov and co-workers [29] have investigated and proposed an anhydrous and a hydrated cocrystal form of FLZ with p-hydroxybenzoic acid, in the latter of which water molecules exert considerable effects on the stabilization of the hydrated form by lowering the lattice energy and formation energy. It deserves attention that cocrystal screening of FLZ by Surov et al. with gallic acid (denoted as 345THB in this study), 3,5-dihydroxybenzoic acid (denoted as 35DHB) has proved to be unsuccessful by employing acetonitrile, methanol, and water as solvents, while cocrystal hydrates were produced by using acetone solvent in this study, which leads us to the conclusion that, in addition to crystal engineering considerations, factors such as solvent of crystallization play a significantly important role in the preparation of cocrystals. Experimental cocrystallization verification of FLZ with a series of hydroxybenzoic acids has resulted in five new molecular assemblies.

### 3.2. Crystallographic Analysis

Among five hydrates, three forms are in a 1:1:1 ratio and two forms are in a 1:0.5:1 ratio. The nature of the above complexes has been confirmed by the geometry of the carboxyl group and the carboxylic acid proton location [34], which revealed that no proton transfer occurred in these five structures, so all these five solid forms could be defined as cocrystals. Noticeably, water molecules play a significant role in the formation and stabilization of the crystal structures, acting both as the hydrogen bond acceptor and donor. The crystallographic parameters for the title compounds are summarized in Table 1, and hydrogen bonding information is presented in Appendix A. CCDC 2203456, 2203457, 2203458, 2203459, and 2203460 contain the supplementary crystallographic data for FLZ−24DHB−H_2_O (1:0.5:1, S1), FLZ−34DHB−H_2_O (form I, 1:0.5:1, S2; form II, 1:1:1, S3), FLZ−35DHB−H_2_O (1:1:1, S4), FLZ−345THB−H_2_O (1:1:1, S5), respectively.

FLZ–34DHB–H_2_O (1:0.5:1) belongs to the monoclinic crystal system, space group *P* (Z = 2). Each asymmetric unit contains one FLZ, a half symmetry independent 24DHB, and one water molecule (Z′ = 1). The 24DHB molecule is disordered over two positions with 50% occupancy. An intramolecular O_3_−H_3_ . . . O_4_ hydrogen bond could be observed within the 24DHB molecule. As is shown in Figure 2a, the crystal packing in the structure is dominantly governed by robust hydrogen bonds associated with the hydroxyl and carbonyl hydroxyl groups of the 24DHB molecule as donors and triazolyl N_3_ atom of the FLZ molecule as acceptors (O_5_^a^_a_−H_5_^a^ . . . N_3_, O_2_^a^−H_2_^a^ . . . N_3_). Besides, serving as a bridge, water molecules connect with the carbonyl group of 24DHB molecules via O−H . . . O hydrogen bonds acting as hydrogen donors, simultaneously connecting the hydroxyl group of the FLZ molecule as hydrogen-acceptors (Figure 2a). These main intermolecular interactions lead to a centrosymmetric system composed of four FLZ molecules, two 24DHB, and four water molecules (Figure 2b). The centrosymmetric systems are further connected into infinite chains stabilized by C_4_−H_4_ . . . F_1_ hydrogen bonds that emerged between FZL molecules as well as other weak contacts (Figure 2c).

FLZ–34DHB–H_2_O (1:0.5:1). Cocrystallization of FLZ and 34DHB could generate two cocrystal hydrates in different stoichiometry: FLZ−34DHB−H_2_O (1:1:1) and FLZ−34DHB−H_2_O (1:0.5:1), the former of which extremely assemble that of FLZ–24DHB–H_2_O (1:0.5:1), belonging to the triclinic system, space group *P* (*Z* = 2) with one FLZ, a half symmetry independent 34DHB, and one water molecule in the asymmetric unit (Z′ = 1, Figure 3a). The 34DHB molecule is disordered over two positions with 50% occupancy. The structure is governed by O_3_^a^−H_3_^a^ . . . N_3_ hydrogen bonding interaction associated with the triazolyl N_3_ atom of FLZ and the carbonyl hydroxyl group of 34DHB (Figure 3b). The water molecules connect with the FLZ molecule via O_1_−H_1_ . . . O_6_ and O_6_−H_6b_ . . . N_6_ hydrogen bonds and interact with 34DHB via O_6_−H_6a_ . . . O_2_ and O_6_−H_6a_ . . . O_5_ hydrogen bonds at the same time (Figure 3b). These main intermolecular interactions result in a centrosymmetric unit consisting of four FLZ molecules, two 34DHB, and four water molecules (Figure 3b). The centrosymmetric systems are further connected into infinite chains viewed along crystallographic a-axis, stabilized by C_8_−H_8_ . . . F_1_ hydrogen bonds emerged between FZL molecules as well as other weak contacts (Figure 3c). The main packing pattern of FLZ−34DHB−H_2_O (1:0.5:1) is similar to that of the crystal structure of FLZ−24DHB−H_2_O (1:0.5:1) which is likely due to similar structure and functional groups of the two acid ligands.

FLZ–34DHB–H_2_O (1:1:1) belongs to the monoclinic system, space group *P*2_1_ (Z = 2), with the asymmetric unit containing one molecule each of FLZ, 34DHB, and water (Z′ = 1, Figure 4a). The FLZ molecule interacts with 34DHB and water molecules via O_5_−H_5_ . . . N_6_ and O_6_−H_6B_ . . . N_2_ hydrogen bonds (Figure 4a). An intermolecular hydrogen bond (O_4_−H_4_ . . . N_3_) associated with the hydroxyl groups of 34DHB and the triazolyl N_3_ atom of FLZ, hydrogen bonds (O_6_−H_6A_ . . . O_4_, O_6_−H_6B_ . . . N_2_) constituted by water molecules, together with the O_1_−H_1_ . . . O_2_ hydrogen bond formed between the carbonyl group of 34DHB and the hydroxyl group of FLZ resulted in the formation of a six-member cyclic unit composed of two FLZ molecules, two 34DHB, and two water molecules (Figure 4a,b). Such aggregates are further connected into layers viewed down the crystallographic a-axis, stabilized by the “linkage” water molecules and other weak contacts (Figure 4c). Two cocrystal polymorphic forms of FLZ−34DHB−H_2_O (1:0.5:1) and FLZ−34DHB−H_2_O (1:1:1) present distinct crystal arrangements relative to their molecular conformation of FLZ and basic building blocks (Figure 4b,c).

FLZ–35DHB–H_2_O (1:1:1) belongs to the monoclinic system, space group *P*2_1_/*n* (Z = 4) with the asymmetric unit containing one molecule each of FLZ, 35DHB, and water (Z′ = 1, Figure 5a). The FLZ molecule interacts with 35DHB and the water molecule through intermolecular O_2_−H_2_ . . . N_6_ and O_1_−H_1_ . . . O_6_ hydrogen bonds (Figure 5a). In addition, the water molecule is attached to the carbonyl/hydroxyl groups of adjacent 35DHB molecules through O_6_−H_6A_ . . . O_2_ and O_6_−H_6B_ . . . O_5_ hydrogen bonds (Figure 5b), resulting in the formation of a 3D structure by the aid of O_1_−H_1_ . . . O_6_ hydrogen bond involved with FLZ and 35DHB (Figure 5c).

FLZ–345THB–H_2_O (1:1:1) belongs to the monoclinic system, space group *P*2_1_/*c* (Z = 4), with the asymmetric unit containing one molecule each of FLZ, 345 THB, and water (Z′ = 1, Figure 6a). In the asymmetric unit, the 345THB molecule interacts with FLZ and water molecules through strong intermolecular O_2_−H_2_ . . . N_6_ and O_6_−H_6_ . . . O_7_ hydrogen bonds, respectively (Figure 6a). The water molecule also interacts with the hydroxyl/carboxyl groups of 345THB molecule via O_7_−H_7A_ . . . O_5_ and O_7_−H_7B_ . . . O_3_ hydrogen bonds as hydrogen donors (Figure 6b). The overall structure features a sandwich-like pattern stabilized by van der Waals and other weak contacts with water and 345THB molecules stacked between double layers of FLZ, viewed along the crystallographic c-axis (Figure 6c). The water molecules incorporated in the crystal lattice may balance the ratio of hydrogen donors and hydrogen acceptors or play a crucial role in the elimination of repulsive interactions between the CCF and API [37,38].

### 3.3. Structure Overlay

Literature reported by Caira, et al. [14] has seen the torsional flexibility in FLZ polymorphs. The two triazole rings of FLZ may be twisted after the introduction of CCFs or water/solvent molecules. To investigate whether the configuration of the FLZ molecule has changed in the newly obtained cocrystal forms, the molecular overlay was performed, and it can be seen that the configuration of FLZ varied greatly among the five cocrystal hydrate forms concerning FLZ form II (Table 2, Figure 7). The formation of the polymorphic form of FLZ−34DHB−H_2_O (1:1:1) enables the most obvious change of FLZ configuration, wherein one of the triazole rings flips, and the corresponding torsion angle τ1 rotates 116.87° in comparison with FLZ form II. Hydrogen-bonding interactions with CCFs can alter the conformation of the molecule due to the requirements of different molecular packing patterns, on the other hand, the conformation flexibility of the API has prompted the molecule to have different conformers such that the different CCFs could recognize, thus extending the scope of cocrystal screening and amplifying the range of solid forms.

### 3.4. PXRD Analysis

Figure 8 presents the experimental and calculated PXRD patterns of the synthesized cocrystals, from 3° to 40° (2*θ*). Herein, the novelty of the individual cocrystals is obvious because they are observed to be different from their respective starting materials. The peaks observed from the experimental PXRD modes happen to coincide with those simulated results calculated from the SCXRD data, suggesting a high purity level of the cocrystal powder samples.

### 3.5. Thermal Analysis

Employed as a powerful tool to determine whether the endotherms corresponded to the loss of solvent/water or a melting event, DSC and TG measurements were performed to trace solvent or water molecules present in the structures [39]. The DSC thermograms of CCFs used in this study have been presented in Appendix A. The hygroscopic nature of FLZ (form II) was evidenced with ~2% mass loss in the range of ~46–120 °C from the TG curve (Figure 9b), corresponding to the minor endothermic peak at 78.76 °C in the DSC profile (Figure 9a), while the second endothermic peak at 141.00 °C can be attributed to the melting event (Figure 9a). Interestingly, except for FLZ−35DHB−H_2_O (1:1:1), the signals of the other four hydrates at ~50–180 °C associated with the release of the water molecules are involved in a relatively strong endothermic effect at the same time.

The thermal data analysis for FLZ−24DHB−H_2_O (1:0.5:1) indicates that a two-step of total ~3.78% (cal. 4.49%) mass loss to dehydration occurred in the ~68–108 °C and ~108–146 °C temperature range, which coincided with the endothermic peak at 98.34 °C in the DSC thermogram. Another broad and weak endothermic peak was generated at 176.67 °C, which is attributed to the decomposition of the compound.

The FLZ−34DHB−H_2_O (1:0.5:1) cocrystal presents a mass loss of ~3.50% (cal. 4.49%) to dehydration in the range of ~56–100 °C and ~100–156 °C and an endothermic peak at 92.27 °C taken as the melting process of the sample. The TG curve of FLZ−34DHB−H_2_O (1:1:1) shows a mass loss of ~3.76% (cal. 3.77%) to dehydration in the range of ~54–108 °C and ~108–160 °C, supported by the endothermic peak at 103.72 °C in DSC curve.

In the case of FLZ−35DHB−H_2_O (1:1:1), the presence of water is also evident from the DSC measurement. The signal at 110.82 °C is ascribed to the release of the water molecules from the crystal lattice, a phase transition peak where water molecules are evaporating, supported by the weight loss of ~3.45% (cal. 3.77%) in the ~56–124 °C temperature range in the TG curve, which is followed by two endothermic peaks at 145.12 °C and 173.71 °C, ascribed to the melting process of dehydrated FLZ−35DHB−H_2_O (1:1:1) form.

The TG curve for FLZ−345THB−H_2_O (1:1:1) shows a mass loss of ~3.57% (cal. 3.64%) in the ~84–128 °C and ~128–180 °C range, which can be attributed to the release of water molecules. The DSC curve for the FLZ−345THB−H_2_O (1:1:1) shows a strong endothermic peak at 108.60 °C, related to the dehydration/melting point of this compound.

### 3.6. FT−IR Analysis

FT−IR analysis was conducted to identify the noncovalent interactions within the obtained multi-component crystals. To our knowledge, the –C=O and –OH functional groups possess the most sensitive IR bands, which show pronounced shifts in the spectra of the hydrogen-bonded complexes, helping distinguish cocrystal formation. Generally, the characteristic absorption peaks of the –OH and –C=O groups are in the range of 3300–3600 cm^−1^ and 1740–1600 cm^−1^, respectively. The results of the vibrational analysis of FLZ and the multi-component crystals have been given in Table 3 and Appendix A. In the present study, the IR band observed at 3117 cm^−1^ has been assigned to the −OH stretching vibration in 2,4-difluoro phenol of FLZ, and the deviation in comparison with literature is most likely to be due to the presence of a strong intramolecular hydrogen bond [40]. These spectral comparisons of the synthesized cocrystals and the individual starting materials reflect that the hydroxyl and carbonyl groups of acids participate in hydrogen-bonding interactions accompanying cocrystal formation.

### 3.7. Stability Study

Being a hygroscopic phase, FLZ is unstable under high humidity circumstances, which would easily transform into FLZ monohydrate (CSD refcode IVUQIZ, Figure 10a), while FLZ monohydrate converted to FLZ form II under the storage of the high-temperature condition for 5 days (Figure 10b). Water molecules interact with hydroxyl O atom and triazole N_3_ and N_6_ atoms of FLZ via O−H . . . O and O−H . . . N hydrogen bonds in the form of FLZ monohydrate (Appendix A). It is worth noting that obtained cocrystal hydrates show impressive stability properties against high-humidity accelerated conditions, which was reasonable because CCFs and water molecules have occupied the main hydrogen bonding sites of FLZ, significantly reducing the probability of water molecules approaching FLZ. In particular, FLZ−34DHB−H_2_O (1:1:1, Figure 10e) is stable against high temperature, high humidity, and illumination, enabling it to be a good candidate. Interestingly, FLZ−24DHB−H_2_O (1:0.5:1, Figure 10c) and FLZ−34DHB−H_2_O (1:0.5:1, Figure 10d) converted to the physical mixture of amorphous and crystalline phases, while in the case of FLZ−35DHB−H_2_O (1:1:1, Figure 10f), water molecules escaped from the crystal lattice, thus forming an anhydrous form after storage in high-temperature condition for 5 days (Figure 10).

As is revealed by DSC, the processes of dehydration and melting took place at the same time in FLZ−24DHB−H_2_O (1:0.5:1) and FLZ−34DHB−H_2_O (1:0.5:1), while dehydration point is below that of melting point in FLZ−35DHB−H_2_O (1:1:1), so it is reasonable that crystal collapse and, hence, amorphism is evident in the former two structures followed by the escape of water molecules, and appearance of the dehydrated cocrystal form is observed in the latter one. It is obvious that by employing crystal engineering, the hygroscopicity associated with commercialized FLZ form II could be significantly reduced, and, hence, the physical stability of parent API could be modulated effectively, which could be attributed to the rearrangements of the crystal lattice and intermolecular interactions after cocrystallization.

### 3.8. Apparent Powder and Intrinsic Dissolution Experiments

The apparent powder dissolution rate can be used to assess the kinetic dissolution profile of the bulk drug and cocrystals, which varies with time during the cocrystal dissolution process. The kinetic solubility is affected by the surface area, phase transformations, particle size, drug distribution, fluid dynamics as well as experimental conditions, and other parameters [41,42]. The IDR determines the dissolution dynamics of compounds at a constant temperature and surface area, the rate at which equilibrium solubility is reached [43], which helps to approximately simulate the in vivo dissolution performances of drug formulation [44]. The dissolution profiles of FLZ form II and its three cocrystal hydrates in pure water were evaluated (Figure 11), with exception of FLZ−34DHB−H_2_O (1:0.5:1) and FLZ−345THB−H_2_O (1:1:1), since cocrystal powders of these two forms could not be prepared on a large-scale. An increase in initial powder dissolution rate was observed in obtained cocrystals, and FLZ−35DHB−H_2_O (1:1:1) showed a higher IDR compared with FLZ form II, which is advantageous for the development of immediate-release drug formulations. As evidenced by the experimental phenomenon, cocrystals disperse rapidly from the basket into the dissolution medium, while the FLZ pure drug was agglomerated. The reasons for the improved dissolution rate could be ascribed to the high solubility of CCFs and the dissociation of cocrystals. Both the order of apparent and intrinsic dissolution rate of FLZ−35DHB−H_2_O (1:1:1) > FLZ−34DHB−H_2_O (1:1:1) > FLZ−24DHB−H_2_O (1:0.5:1) show a good correlation with the aqueous solubility order of CCFs in water at 25 °C: 35DHB (84 g L^−1^) > 34DHB (29 g L^−1^) > 24DHB (8 g L^−1^).

## 4. Conclusions

In this paper, five new FLZ cocrystal hydrates with a series of hydroxybenzoic acids, namely, 24DHB, 34DHB, 35DHB, and 345THB were obtained by solvent-assisted grinding and slow evaporation methods, extending the range of FLZ solid forms at large. Crystals of these five multi-components were obtained and completely elucidated by SCXRD, which demonstrated that robust (carboxylic acid/hydroxyl) O−H . . . N (triazolyl) heterosynthon is responsible for self-assemblies in FLZ with hydroxybenzoic acids. Water molecules play the role of supramolecular “linkage” in the strengthening and stabilization of these hydrates by participating in the creation of O−H . . . O hydrogen bonds interacting with FLZ and acids.

In particular, FLZ−34DHB−H_2_O (1:1:1) significantly reduces hygroscopicity and hence improves the stability of FLZ, which was reasonable because CCFs and water molecules have occupied the main hydrogen bonding sites of FLZ. The accelerated stability test results also suggest that water molecules appear to contribute noticeably to the stabilization of the resulting crystal net. An increased initial powder dissolution rate was observed in obtained cocrystals, and FLZ−35DHB−H_2_O (1:1:1) showed a higher IDR compared with FLZ form II, which is advantageous for the development of immediate-release drug formulations. The order of apparent and intrinsic dissolution rate of cocrystals show a good correlation with the aqueous solubility order of CCFs in water at 25 °C. An excellent stability property and improved dissolution rate of FLZ−34DHB−H_2_O (1:1:1) enable it to be a good candidate for further pharmaceutical development.

Our study emphasizes the benefit of crystal engineering applied to modify the stability and dissolution rate of the commercialized FLZ form II. The design and preparation of these new pharmaceutical derivatives also give a better understanding and knowledge of the synthonic hierarchy in organic cocrystals. The availability of these crystal structures reported herein sheds light on the nature of the molecular association of FLZ with hydroxybenzoic acids, together with a recommendation for the cocrystal design and synthesis of nitrogen heterocycle compounds.

## Figures and Tables

**Figure 1 pharmaceutics-14-02486-f001:**
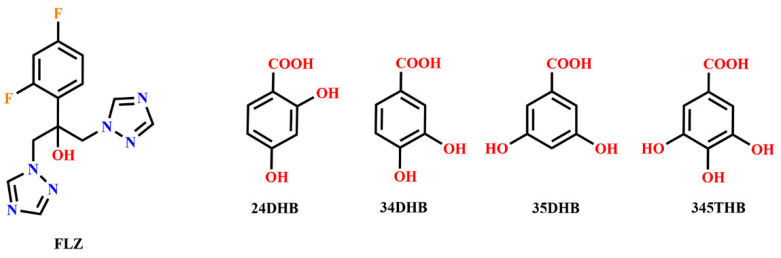
Molecular diagram of FLZ and the phenolic acids used in this study.

**Figure 2 pharmaceutics-14-02486-f002:**
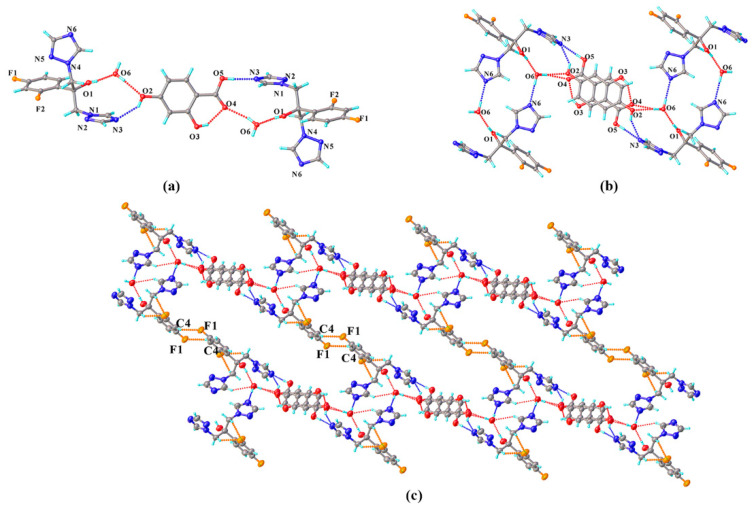
(**a**) Hydrogen-bonding interactions in two asymmetric units; (**b**) the centrosymmetric system composed of four FLZ molecules, two 24DHB, and four water molecules; (**c**) hydrogen-bonded chains stabilized by C_4_−H_4_ . . . F_1_ hydrogen bonds and other weak contacts.

**Figure 3 pharmaceutics-14-02486-f003:**
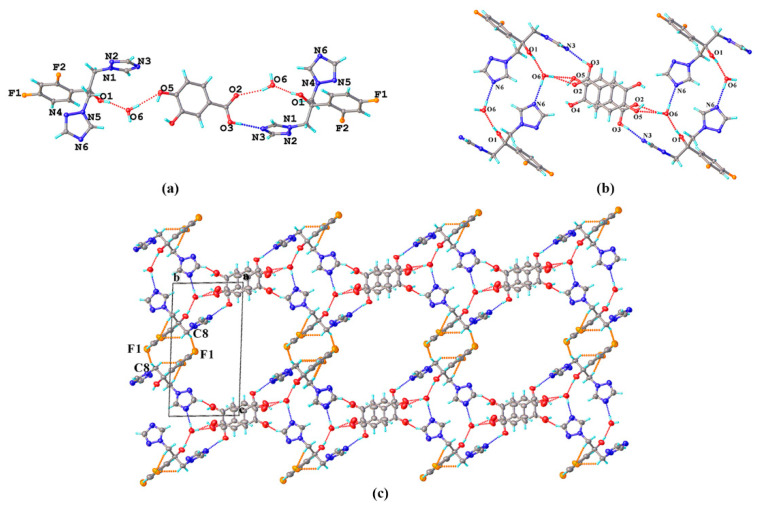
(**a**) Hydrogen-bonding interactions in two asymmetric units; (**b**) the centrosymmetric system composed of four FLZ molecules, two 34DHB, and four water molecules; (**c**) hydrogen-bonded chains stabilized by C_8_−H_8_ . . . F_1_ hydrogen bond and other weak contacts viewed along a-axis. a,b,c in subfigure (**c**) represents the three axes of the cell.

**Figure 4 pharmaceutics-14-02486-f004:**
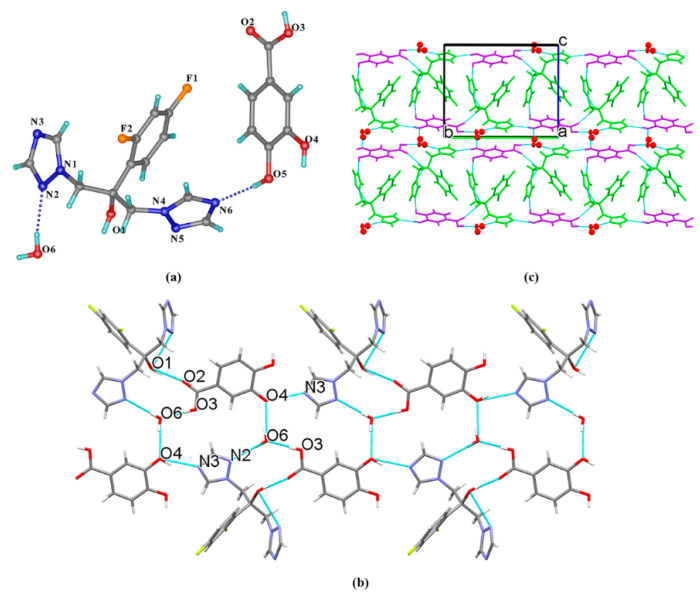
(**a**) Hydrogen-bonding interactions in an asymmetric unit; (**b**) the 2D hydrogen-bonded motifs six-member cyclic unit composed of FLZ, 34DHB, and water molecules; (**c**) hydrogen-bonded 3D structure stabilized by water molecules and other weak contacts with water molecules in space-filled mode viewed down the crystallographic a-axis. a,b,c in subfigure (**c**) represents the three axes of the cell.

**Figure 5 pharmaceutics-14-02486-f005:**
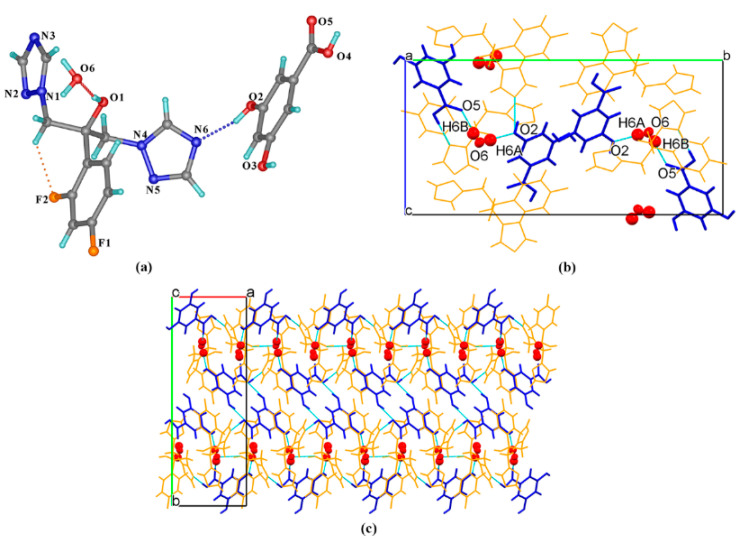
(**a**) Hydrogen-bonding interactions in an asymmetric unit; (**b**) interactions between FLZ, 35DHB, and water molecules; (**c**) 3D structure viewed along the c-axis with water molecules in space-filled mode, FLZ in wire-framed mode. a,b,c in subfigure (**b**) and (**c**) represents the three axes of the cell.

**Figure 6 pharmaceutics-14-02486-f006:**
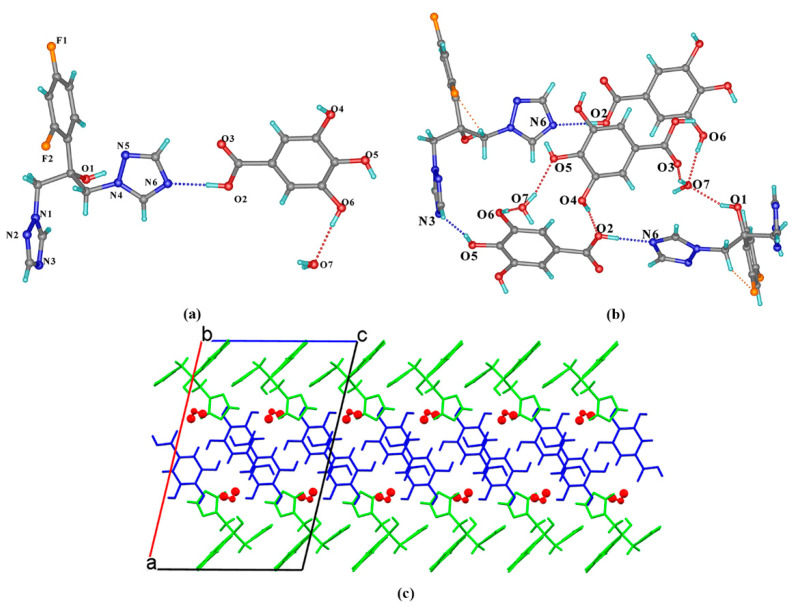
(**a**) Hydrogen-bonding interactions in an asymmetric unit; (**b**) the hydrogen-bonded motifs composed of FLZ, 345THB, and water molecules; (**c**) hydrogen-bonded sandwich-like 3D structure with water molecules in space-filled mode. a,b,c in subfigure (**c**) represents the three axes of the cell.

**Figure 7 pharmaceutics-14-02486-f007:**
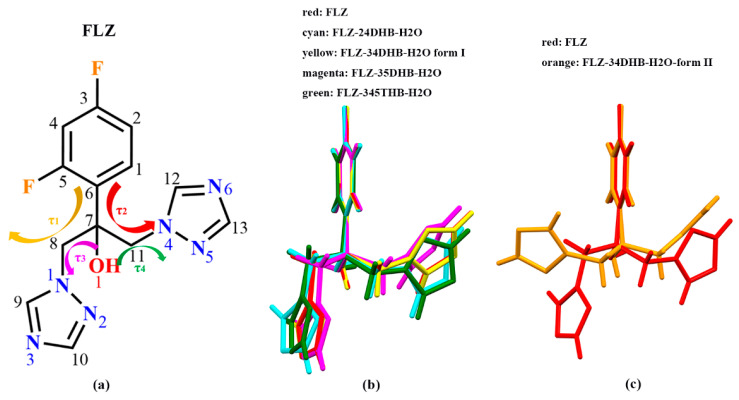
(**a**) Molecular diagram of fluconazole (FLZ). Four torsion angles: τ1 = C_6_−C_7_−C_8_−N_1_, τ2 = C_6_−C_7_−C_11_−N_4_, τ3 = O_1_−C_7_−C_8_−N_1_, τ4 = O_1_−C_7_−C_11_−N_4_. (**b**,**c**) Structure Overlay of FLZ conformations in the FLZ pure drug (form II) and its cocrystal hydrate forms.

**Figure 8 pharmaceutics-14-02486-f008:**
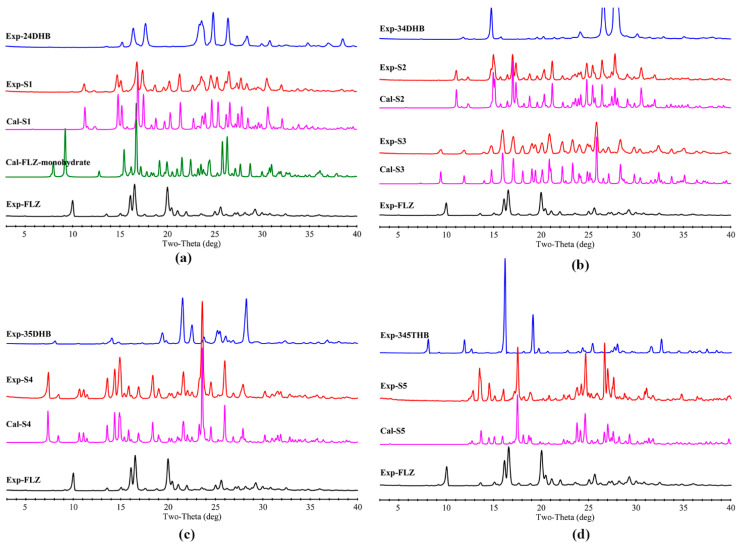
Comparison of experimental FLZ used in this study, simulated FLZ monohydrate, and the experimental and simulated PXRD patterns of (**a**) FLZ−24DHB−H_2_O (1:0.5:1, S1), (**b**) FLZ−34DHB−H_2_O (1:0.5:1, S2), FLZ−34DHB−H_2_O (1:1:1, S3), (**c**) FLZ−35DHB−H_2_O (1:1:1, S4), (**d**) FLZ−345THB−H_2_O (1:1:1, S5).

**Figure 9 pharmaceutics-14-02486-f009:**
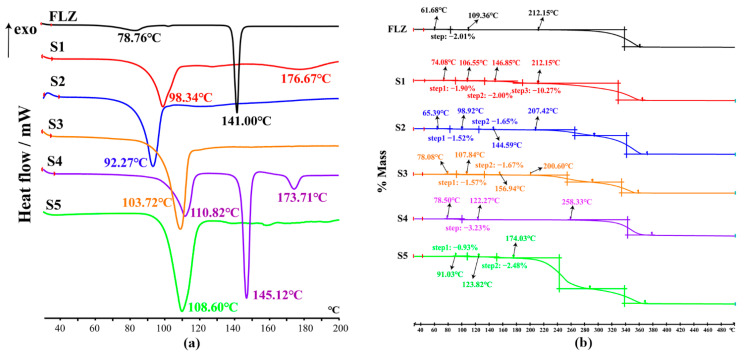
Thermal analysis curve for FLZ raw material, (S1) FLZ−24DHB−H_2_O (1:0.5:1), (S2) FLZ−34DHB−H_2_O (1:0.5:1), (S3) FLZ−34DHB−H_2_O (1:1:1), (S4) FLZ−35DHB−H_2_O (1:1:1), (S5) FLZ−345THB−H_2_O (1:1:1), indicating their dehydration, melting and decomposition events. (**a**) DSC thermograms, (**b**) TG profiles.

**Figure 10 pharmaceutics-14-02486-f010:**
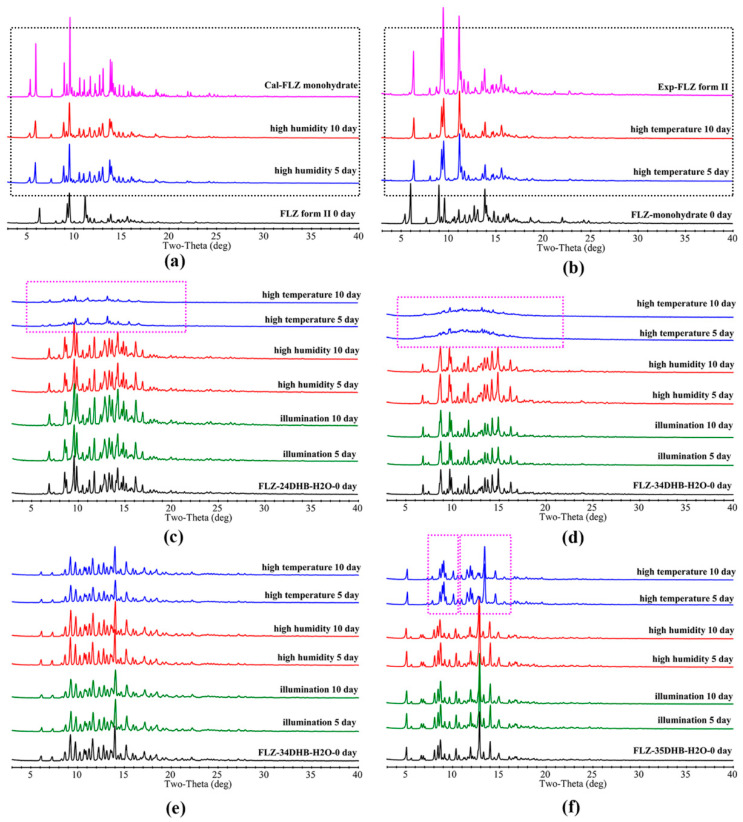
Accelerated stability results of (**a**) FLZ, (**b**) FLZ monohydrate, (**c**) FLZ−24DHB−H_2_O (1:0.5:1), (**d**) FLZ−34DHB−H_2_O (1:0.5:1), (**e**) FLZ−34DHB−H_2_O (1:1:1), (**f**) FLZ−35DHB−H_2_O (1:1:1). The dotted box in the figure highlights peaks of transformed phases.

**Figure 11 pharmaceutics-14-02486-f011:**
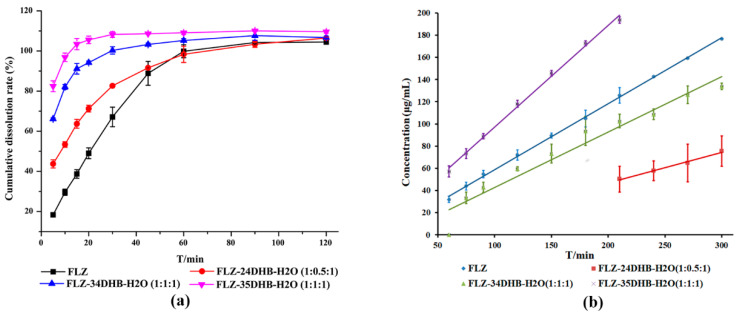
(**a**) Powder dissolution studies, (**b**) intrinsic dissolution rates of marketed FLZ form II and its cocrystal hydrates in ultrapure water.

**Table 1 pharmaceutics-14-02486-t001:** Crystallographic data and structure refinement details.

Title 1	FLZ−24DHB−H_2_O	FLZ−34DHB−H_2_O	FLZ−34DHB−H_2_O	FLZ−34DHB−H_2_O	FLZ−34DHB−H_2_O
Formula	C_16.5_H_17_F_2_N_6_O_4_	C_16.5_H_17_F_2_N_6_O_4_	C_20_H_20_F_2_N_6_O_6_	C_20_H_20_F_2_N_6_O_6_	C_20_H_20_F_2_N_6_O_7_
Formula weight	401.36	401.36	478.42	478.42	494.42
Crystal size (mm)	0.16 × 0.28 × 0.47	0.21 × 0.28 × 0.39	0.17 × 0.26 × 0.38	0.24 × 0.28 × 0.45	0.03 × 0.18 × 0.39
Description	block	block	plate	column	column
Crystal system	triclinic	triclinic	monoclinic	monoclinic	monoclinic
Space group	*P*−1 *P*	*P*	*P*2_1_	*P*2_1_/*n*	*P*2_1_/*c*
Unit cell parameters (Å, °)	7.542 (1)	7.475 (1)	5.970 (1)	8.497 (1)	21.104 (1)
	7.960 (1)	8.105 (1)	14.762 (1)	23.250 (1)	7.425 (1)
	15.683 (1)	15.619 (1)	11.971 (1)	11.632 (1)	13.951 (1)
	90.51 (1)	85.10 (1)	90	90	90
	103.54 (1)	76.48 (1)	94.40 (1)	103.23 (1)	103.25 (1)
	101.02 (1)	79.33 (1)	90	90	90
Volume (Å^3^)	897.02 (5)	903.22 (6)	1051.85 (2)	2236.91 (4)	2128.13 (4)
*Z*/*Z’*	2/1	2/1	2/1	4/1	4/1
Density (g cm^−3^)	1.486	1.476	1.511	1.421	1.543
Independent reflections	3413	3427	3872	4369	4022
Reflections with *I* > 2*σ*(*I*)	3220	3245	3855	4028	3931
*R_int_*	0.029	0.016	0.033	0.020	0.029
final *R*, *wR*(*F*^2^) value	0.041, 0.110	0.050, 0.145	0.029, 0.076	0.036, 0.094	0.041, 0.106
GOF	1.092	1.059	1.042	1.059	1.009
Completeness	0.980	0.975	0.989	0.998	0.981
CCDC	2203456	2203457	2203458	2203459	2203460

**Table 2 pharmaceutics-14-02486-t002:** Torsion angles of FLZ and its cocrystal hydrates.

Samples	τ1(°)	τ2(°)	τ3(°)	τ4(°)
FLZ	174.77	68.20	56.74	57.75
FLZ−24DHB−H_2_O (1:0.5:1)	−177.03 (10)	−55.20 (12)	67.01 (13)	64.67 (13)
FLZ−34DHB−H_2_O (1:0.5:1)	−177.76 (13)	−56.10 (17)	66.17 (18)	64.22 (18)
FLZ−34DHB−H_2_O (1:1:1)	57.90 (1)	−45.60 (1)	−61.90 (1)	72.90 (1)
FLZ−35DHB−H_2_O (1:1:1)	179.41 (10)	−59.37 (12)	56.32 (14)	61.31 (12)
FLZ−345THB−H_2_O (1:1:1)	−173.17 (10)	−64.57 (14)	68.88 (14)	62.19 (14)

**Table 3 pharmaceutics-14-02486-t003:** FT−IR spectral frequencies of FLZ cocrystal hydrates (in cm^−1^).

Compound	2,4-Difluoro Phenol	Carbonyl Group
	–O−H Stretch	–C=O Strech
FLZ form II	3117	-
FLZ−24DHB−H_2_O (1:0.5:1)	3512, 3110	1619
FLZ−34DHB−H_2_O (1:0.5:1)	3526, 3112	1668 (w)
FLZ−34DHB−H_2_O (1:1:1)	3439	1594
FLZ−35DHB−H_2_O (1:1:1)	3472	1681
FLZ−345THB−H_2_O (1:1:1)	3453, 3339, 3121	1679

## Data Availability

CCDC 2203456–2203460 contain the supplementary crystallographic data for this paper. The data can be obtained free of charge via http://www.ccdc.cam.ac.uk/conts/retrieving.html (accessed on 1 September 2022) or by emailing data_request@ccdc.cam.ac.uk or by contacting The Cambridge Crystallographic Data Centre, 12 Union Road, Cambridge CB2 1EZ, UK; Fax: +44-1223-336033.

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
