# Peer review of "Design, Preparation, Characterization and Evaluation of Five Cocrystal Hydrates of Fluconazole with Hydroxybenzoic Acids"

_pharmaceutics, 2022, doi:10.3390/pharmaceutics14112486_

Round 1

Reviewer 1 Report

The authors obtain and characterize five cocrytal hydrates of fluconazole. Minor comments for the manuscript.

1. Detail the preparation of single crystal growth. 

2. Please investigate and explain why the powder dissolution of FLZ is the slowest while it shows relative higher dissolution in IDR? 

Reviewer 2 Report

In this study, crystal engineering is applied to modify the dissolution rate and stability of FLZ form II. It also clarifies the nature of the molecular association of FLZ with hydroxybenzoic acids, suggesting novel pharmaceutical derivatives of FLZ. Overall, the paper is well-written, and the conclusions are supported by data. Below are a few minor comments.

LINES 66-88: I would not anticipate the results in the introductory section and would expand the part relating to the motivations of the research and the impact /application of the results obtained.

LINES 112-114: Could the authors be more precise on the specifications of the conditions used (range of temperatures, stirring or stationary conditions, etc..).

LINE 137: Why has the lid been perforated? Please add a comment. Have the dsc runs been conducted under a dry nitrogen gas flow?

Reviewer 3 Report

In recent years the area of cocrystals has been most widely investigated to improve the stability or solubility of drug substances. The current research involving the development of five different cocrystals of Fluconazole is interesting. The manuscript was well written with a good flow of content, and the results were well explained with proper justifications and supporting literature. The current manuscript needs to be revised for a few of the below-mentioned comments:

  1. Within the introduction section of the manuscript, please include any challenges associated with Fluconazole and provide a small note for choosing the drug candidate.
  2. Please make a note of the main scope of the research work. 
  3. Were the formulations of liquid-assisted grinding dried to remove any trace amount of solvent?
  4. Please do not discuss the unsuccessful trials for cocrystal formation in the methodology section of the manuscript.
  5. Provide the information for the maximum temperature to which the samples were heated in DSC.
  6. Please update the headings of sections 2.7 and 2.8.
  7. Why was 600 mL of dissolution media chosen?
  8. Provide information for the calibration curve.
  9. Please provide the DSC thermal scans of five coformers employed in the current investigation.
  10. The conclusion is very lengthy and needs to be revised. Please focus only on the main findings and outcomes of the current research.
